# Mutual Information Estimation via $f$-Divergence and Data Derangement Based Learning Models

## Abstract

Estimating mutual information accurately is pivotal across diverse applications, from machine learning to communications and biology, enabling us to gain insights into the inner mechanisms of complex systems. Yet, dealing with high-dimensional data presents a formidable challenge, due to its size and the presence of intricate relationships. Recently proposed neural methods employing variational lower bounds on the mutual information have gained prominence. However, these approaches suffer from either high bias or high variance, as the sample size and the structure of the loss function directly influence the training process. In this paper, we propose a novel class of discriminative mutual information estimators based on the variational representation of the $f$-divergence. We investigate the impact of the permutation function used to obtain the marginal training samples and present a novel architectural solution based on derangements. The proposed estimator is flexible since it exhibits an excellent bias/variance trade-off. The comparison with state-of-the-art neural estimators, through extensive experimentation within established reference scenarios, shows that our approach offers higher accuracy and lower complexity.

## 1 Introduction

The mutual information (MI) between two multivariate random variables, $X$ and $Y$, is a fundamental quantity in statistics, representation learning, information theory, communication engineering and biology (Goldfeld & Greenewald, 2021; Tschannen et al., 2020; Guo et al., 2005; Pluim et al., 2003). It quantifies the statistical dependence between $X$ and $Y$ by measuring the amount of information obtained about $X$ via the observation of $Y$, and it is defined as

$$I(X;Y) = \mathbb{E}_{(\mathbf{x},\mathbf{y}) \sim p_{XY}(\mathbf{x},\mathbf{y})} \left[ \log \frac{p_{XY}(\mathbf{x},\mathbf{y})}{p_X(\mathbf{x})p_Y(\mathbf{y})} \right]. \tag{1}$$

Unfortunately, computing $I(X;Y)$ is challenging since the joint probability density function $p_{XY}(\mathbf{x},\mathbf{y})$ and the marginals $p_X(\mathbf{x}), p_Y(\mathbf{y})$ are usually unknown, especially when dealing with high-dimensional data. Some recent techniques (Papamakarios et al., 2017; Letizia & Tonello, 2022) have demonstrated that neural networks can be leveraged as probability density function estimators and, more in general, are capable of modeling the data dependence. Discriminative approaches (Raina et al., 2003; Tonello & Letizia, 2022) compare samples from both the joint and marginal distributions to directly compute the density ratio (or the log-density ratio)

$$R(\mathbf{x},\mathbf{y}) = \frac{p_{XY}(\mathbf{x},\mathbf{y})}{p_X(\mathbf{x})p_Y(\mathbf{y})}. \tag{2}$$

We focus on discriminative MI estimation since it can in principle enjoy some of the properties of implicit generative models, which are able of directly generating data that belongs to the same distribution of the input data without any explicit density estimate. In this direction, the most successful technique is represented by generative adversarial networks (GANs) (Goodfellow et al., 2014). The adversarial training pushes the discriminator $D(\mathbf{x})$ towards

the optimum value

$$\hat{D}(\mathbf{x}) = \frac{p_{data}(\mathbf{x})}{p_{data}(\mathbf{x}) + p_{gen}(\mathbf{x})} = \frac{1}{1 + \frac{p_{gen}(\mathbf{x})}{p_{data}(\mathbf{x})}}. \tag{3}$$

Therefore, the output of the optimum discriminator is itself a function of the density ratio $p_{gen}/p_{data}$, where $p_{gen}$ and $p_{data}$ are the distributions of the generated and the collected data, respectively.

We generalize the observation of (3) and we propose a family of MI estimators based on the variational lower bound of the $f$-divergence (Poole et al., 2019; Sason & Verdú, 2016). In particular, we argue that the maximization of any $f$-divergence variational lower bound can lead to a MI estimator with excellent bias/variance trade-off.

Since we typically have access only to joint data points $(\mathbf{x}, \mathbf{y}) \sim p_{XY}(\mathbf{x}, \mathbf{y})$, another relevant practical aspect is the sampling strategy to obtain data from the product of marginals $p_X(\mathbf{x})p_Y(\mathbf{y})$, for instance via a shuffling mechanism along $N$ realizations of $Y$. We analyze the impact that the permutation has on the learning and training process and we propose a derangement training strategy that achieves high performance requiring $\Omega(N)$ operations. Simulation results demonstrate that the proposed approach exhibits improved estimations in a multitude of scenarios.

In brief, we can summarize our contributions over the state-of-the-art as follows:

- For any $f$-divergence, we derive a training value function whose maximization leads to a given MI estimator.

- We compare different $f$-divergences and comment on the resulting estimator properties and performance.

- We study the impact of data derangement for the learning model and propose a novel derangement training strategy that overcomes the upper bound on the MI estimation (McAllester & Stratos, 2020), contrarily to what happens when using a random permutation strategy.

- We unify the main discriminative estimators into a publicly available code which can be used to reproduce all the results of this paper.

## 2 Related Work

Traditional approaches for the MI estimation rely on binning, density and kernel estimation (Moon et al., 1995) and $k$-nearest neighbors (Kraskov et al., 2004). Nevertheless, they do not scale to problems involving high-dimensional data as it is the case in modern machine learning applications. Hence, deep neural networks have recently been leveraged to maximize variational lower bounds on the MI (Poole et al., 2019; Nguyen et al., 2010; Belghazi et al., 2018). The expressive power of neural networks has shown promising results in this direction although less is known about the effectiveness of such estimators (Song & Ermon, 2020), especially since they suffer from either high bias or high variance.

Discriminative approaches usually exploit an energy-based variational family of functions to provide a lower bound on the Kullback-Leibler (KL) divergence. As an example, the Donsker-Varadhan dual representation of the KL divergence (Poole et al., 2019; Donsker & Varadhan, 1983) produces an estimate of the MI using the bound optimized by the mutual neural information estimator (MINE) (Belghazi et al., 2018). Another variational lower bound is based on the KL divergence dual representation introduced in (Nguyen et al., 2010) (also referred to as $f$-MINE in (Belghazi et al., 2018)). Both MINE and NWJ suffer from high-variance estimates and to combat such a limitation, the SMILE estimator was introduced in (Song & Ermon, 2020). SMILE is equivalent to MINE in the limit $\tau \to +\infty$. The MI estimator based on contrastive predictive coding (CPC) (van den Oord et al., 2018) provides low variance estimates but it is upper bounded by $\log N$, resulting in a biased estimator. Such upper bound, typical of contrastive learning objectives, has been recently analyzed in the context of skew-divergence estimators (Lee & Shin, 2022).

Another estimator based on a classification task is the neural joint entropy estimator (NJEE) proposed in (Shalev et al., 2022), which estimates the MI as entropies subtraction.

Inspired by the $f$-GAN training objective (Nowozin et al., 2016), in the following, we present a class of discriminative MI estimators based on the $f$-divergence measure. Conversely to what has been proposed so far in the literature, where $f$ is always constrained to be the generator of the KL divergence, we allow for any choice of $f$. Different $f$ functions will have different impact on the training and optimization sides, while on the estimation side, the partition function does not need to be computed, leading to low variance estimators.

## 3 $f$-DIVERGENCE MUTUAL INFORMATION ESTIMATION

The calculation of the MI via a discriminative approach requires the density ratio (2). From (3), we observe that $I(X;Y)$ can be estimated using the optimum GAN discriminator $\hat{D}$ when $p_{data} \equiv p_X p_Y$ and $p_{gen} \equiv p_{XY}$. More in general, the authors in (Nowozin et al., 2016) extended the variational divergence estimation framework presented in (Nguyen et al., 2010) and showed that any $f$-divergence can be used to train GANs. Inspired by such idea, we now argue that also discriminative MI estimators enjoy similar properties if the variational representation of $f$-divergence functionals $D_f(P||Q)$ is adopted.

In detail, let $P$ and $Q$ be absolutely continuous measures w.r.t. $dx$ and assume they possess densities $p$ and $q$, then the $f$-divergence is defined as follows

$$D_f(P||Q) = \int_{\mathcal{X}} q(\mathbf{x}) f\left(\frac{p(\mathbf{x})}{q(\mathbf{x})}\right) d\mathbf{x}, \tag{4}$$

where $\mathcal{X}$ is a compact domain and the function $f : \mathbb{R}_+ \to \mathbb{R}$ is convex, lower semicontinuous and satisfies $f(1) = 0$.

The following theorem introduces $f$-DIME, a class of discriminative mutual information estimators (DIME) based on the variational representation of the $f$-divergence.

**Theorem 1.** *Let $(X, Y) \sim p_{XY}(\mathbf{x}, \mathbf{y})$ be a pair of multivariate random variables. Let $\sigma(\cdot)$ be a permutation function such that $p_{\sigma(Y)}(\sigma(\mathbf{y})|\mathbf{x}) = p_Y(\mathbf{y})$. Let $f^*$ be the Fenchel conjugate of $f : \mathbb{R}_+ \to \mathbb{R}$, a convex lower semicontinuous function that satisfies $f(1) = 0$ with derivative $f'$. If $\mathcal{J}_f(T)$ is a value function defined as*

$$\mathcal{J}_f(T) = \mathbb{E}_{(\mathbf{x},\mathbf{y}) \sim p_{XY}(\mathbf{x},\mathbf{y})}\left[T(\mathbf{x}, \mathbf{y}) - f^*\left(T(\mathbf{x}, \sigma(\mathbf{y}))\right)\right], \tag{5}$$

*then*

$$\hat{T}(\mathbf{x}, \mathbf{y}) = \arg\max_T \mathcal{J}_f(T) = f'\left(\frac{p_{XY}(\mathbf{x}, \mathbf{y})}{p_X(\mathbf{x}) p_Y(\mathbf{y})}\right), \tag{6}$$

*and*

$$I(X;Y) = I_{fDIME}(X;Y) = \mathbb{E}_{(\mathbf{x},\mathbf{y}) \sim p_{XY}(\mathbf{x},\mathbf{y})}\left[\log\left((f^*)'(\hat{T}(\mathbf{x}, \mathbf{y}))\right)\right]. \tag{7}$$

Theorem 1 shows that any value function $\mathcal{J}_f$ of the form in (5), seen as the dual representation of a given $f$-divergence $D_f$, can be maximized to estimate the MI via (7). It is interesting to notice that the proposed class of estimators does not need any evaluation of the partition term.

We propose to parametrize $T(\mathbf{x}, \mathbf{y})$ with a deep neural network $T_\theta$ of parameters $\theta$ and solve with gradient ascent and back-propagation to obtain

$$\hat{\theta} = \arg\max_\theta \mathcal{J}_f(T_\theta). \tag{8}$$

By doing so, it is possible to guarantee that, at every training iteration $n$, the convergence of the $f$-DIME estimator $\hat{I}_{n,fDIME}(X;Y)$ is controlled by the convergence of $T$ towards the tight bound $\hat{T}$ while maximizing $\mathcal{J}_f(T)$, as stated in the following lemma.

**Lemma 1.** *Let the discriminator $T(\cdot)$ be with enough capacity, i.e., in the non parametric limit. Consider the problem*

$$\hat{T} = \arg\max_T \mathcal{J}_f(T) \tag{9}$$

*where $\mathcal{J}_f(T)$ is defined as in (5), and the update rule based on the gradient descent method*

$$T^{(n+1)} = T^{(n)} + \mu \nabla \mathcal{J}_f(T^{(n)}). \tag{10}$$

*If the gradient descent method converges to the global optimum $\hat{T}$, the mutual information estimator defined in (7) converges to the real value of the mutual information $I(X;Y)$.*

The proof of Lemma 1, which is described in the Appendix, provides some theoretical grounding for the behaviour of MI estimators when the training does not converge to the optimal density ratio. Moreover, it also offers insights about the impact of different functions $f$ on the numerical bias.

It is important to remark the difference between the classical variational lower bounds estimators that follow a discriminative approach and the DIME-like estimators. They both achieve the goal through a discriminator network that outputs a function of the density ratio. However, the former models exploit the variational representation of the MI (or the KL) and, at the equilibrium, use the discriminator output directly in one of the value functions reported in Appendix B. The latter, instead, use the variational representation of *any* $f$-divergence to extract the density ratio estimate directly from the discriminator output.

In the upcoming sections, we analyze the variance of $f$-DIME and we propose a training strategy for the implementation of Theorem 1. In our experiments, we consider the cases when $f$ is the generator of: a) the KL divergence; b) the GAN divergence; c) the Hellinger distance squared. Due to space constraints, we report in Sec. A of the Appendix the value functions used for training and the mathematical expressions of the resulting DIME estimators.

## 4 VARIANCE ANALYSIS

In this section, we assume that the ground truth density ratio $\hat{R}(\mathbf{x}, \mathbf{y})$ exists and corresponds to the density ratio in (2). We also assume that the optimum discriminator $\hat{T}(\mathbf{x}, \mathbf{y})$ is known and already obtained (e.g. via a neural network parametrization).

We define $p_{XY}^M(\mathbf{x}, \mathbf{y})$ and $p_X^N(\mathbf{x})p_Y^N(\mathbf{y})$ as the empirical distributions corresponding to $M$ i.i.d. samples from the true joint distribution $p_{XY}$ and to $N$ i.i.d. samples from the product of marginals $p_X p_Y$, respectively. The randomness of the sampling procedure and the batch sizes $M, N$ influence the variance of variational MI estimators. In the following, we prove that under the previous assumptions, $f$-DIME exhibits better performance in terms of variance w.r.t. some variational estimators with a discriminative approach, e.g., MINE and NWJ.

The partition function estimation $\mathbb{E}_{p_X^N p_Y^N}[\hat{R}]$ represents the major issue when dealing with variational MI estimators. Indeed, they comprise the evaluation of two terms (using the given density ratio), and the partition function is the one responsible for the variance growth. The authors in (Song & Ermon, 2020) characterized the variance of both MINE and NWJ estimators, in particular, they proved that the variance scales exponentially with the ground-truth MI $\forall M \in \mathbb{N}$

$$\text{Var}_{p_{XY}, p_X p_Y}\left[I_{NWJ}^{M,N}\right] \geq \frac{e^{I(X;Y)} - 1}{N}$$

$$\lim_{N \to \infty} N \text{Var}_{p_{XY}, p_X p_Y}\left[I_{MINE}^{M,N}\right] \geq e^{I(X;Y)} - 1, \tag{11}$$

where

$$I_{NWJ}^{M,N} := \mathbb{E}_{p_{XY}^M}[\log \hat{R} + 1] - \mathbb{E}_{p_X^N p_Y^N}[\hat{R}]$$

$$I_{MINE}^{M,N} := \mathbb{E}_{p_{XY}^M}[\log \hat{R}] - \log \mathbb{E}_{p_X^N p_Y^N}[\hat{R}]. \tag{12}$$

To reduce the impact of the partition function on the variance, the authors of (Song & Ermon, 2020) also proposed to clip the density ratio between $e^{-\tau}$ and $e^{\tau}$ leading to an estimator (SMILE) with bounded partition variance. However, also the variance of the log-density ratio $\mathbb{E}_{p_{XY}^M}[\log \hat{R}]$ influences the variance of the variational estimators, since it is clear that

$$\mathrm{Var}_{p_{XY}, p_X p_Y}\left[I_{VLB}^{M,N}\right] \geq \mathrm{Var}_{p_{XY}}\left[\mathbb{E}_{p_{XY}^M}[\log \hat{R}]\right], \tag{13}$$

a result that holds for any type of MI estimator based on a variational lower bound (VLB).

The great advantage of $f$-DIME is to avoid the partition function estimation step, significantly reducing the variance of the estimator. Under the same initial assumptions, from (13) we can immediately conclude that

$$\mathrm{Var}_{p_{XY}}\left[I_{fDIME}^M\right] \leq \mathrm{Var}_{p_{XY}, p_X p_Y}\left[I_{VLB}^{M,N}\right], \tag{14}$$

where

$$I_{fDIME}^M := \mathbb{E}_{p_{XY}^M}[\log \hat{R}] \tag{15}$$

is the Monte Carlo implementation of $f$-DIME. Hence, the $f$-DIME class of models has lower variance than any VLB based estimator (MINE, NWJ, SMILE, etc.). The following Lemma provides an upper bound on the variance of the $f$-DIME estimator. Notice that such result holds for any type of value function $\mathcal{J}_f$, so it is not restrictive to the KL divergence.

**Lemma 2.** *Let $\hat{R} = p_{XY}(\mathbf{x}, \mathbf{y})/(p_X(\mathbf{x}) p_Y(\mathbf{y}))$ be the density ratio and assume $Var_{p_{XY}}[\log \hat{R}]$ exists. Let $p_{XY}^M$ be the empirical distribution of $M$ i.i.d. samples from $p_{XY}$ and let $\mathbb{E}_{p_{XY}^M}$ denote the sample average over $p_{XY}^M$. Then, under the randomness of the sampling procedure it follows that*

$$Var_{p_{XY}}\left[\mathbb{E}_{p_{XY}^M}[\log \hat{R}]\right] \leq \frac{4H^2(p_{XY}, p_X p_Y)\left\|\frac{p_{XY}}{p_X p_Y}\right\|_{\infty} - I^2(X; Y)}{M} \tag{16}$$

*where $H^2$ is the Hellinger distance squared.*

Lemma 4, also in the Appendix, characterizes the variance of the estimator in (15) when $X$ and $Y$ are correlated Gaussian random variables. We found out that the variance is finite and we use this result to verify in the experiments that the variance of $f$-DIME does not diverge for high values of MI.

## 5 DERANGEMENT STRATEGY

The discriminative approach essentially compares expectations over both joint $(\mathbf{x}, \mathbf{y}) \sim p_{XY}$ and marginal $(\mathbf{x}, \mathbf{y}) \sim p_X p_Y$ data points. Practically, we have access only to $N$ realizations of the joint distribution $p_{XY}$ and to obtain $N$ marginal samples of $p_X p_Y$ from $p_{XY}$ a shuffling mechanism for the realizations of $Y$ is typically deployed. A general result in (McAllester & Stratos, 2020) shows that failing to sample from the correct marginal distribution would lead to an upper bounded MI estimator.

We study the structure that the permutation law $\sigma(\cdot)$ in Theorem 1 needs to have when numerically implemented. In particular, we now prove that a naive permutation over the realizations of $Y$ results in an incorrect VLB of the $f$-divergence, causing the MI estimator to be bounded by $\log(N)$, where $N$ is the batch size. To solve this issue, we propose a derangement strategy.

Let the data points $(\mathbf{x}, \mathbf{y}) \sim p_{XY}$ be $N$ pairs $(\mathbf{x}_i, \mathbf{y}_i), \forall i \in \{1, \ldots, N\}$. The naive permutation of $\mathbf{y}$, denoted as $\pi(\mathbf{y})$, leads to $N$ new random pairs $(\mathbf{x}_i, \mathbf{y}_j), \forall i$ and $j \in \{1, \cdots, N\}$. The idea is that a random naive permutation may lead to at least one pair $(\mathbf{x}_k, \mathbf{y}_k)$, with $k \in \{1, \ldots, N\}$, which is actually a sample from the joint distribution. Viceversa, the derangement of $\mathbf{y}$, denoted as $\sigma(\mathbf{y})$, leads to $N$ new random pairs $(\mathbf{x}_i, \mathbf{y}_j)$ such that $i \neq j, \forall i$ and $j \in \{1, \cdots, N\}$. Such pairs $(\mathbf{x}_i, \mathbf{y}_j), i \neq j$ can effectively be considered samples from $p_X(\mathbf{x}) p_Y(\mathbf{y})$. An example using these definitions is provided in Appendix D.1.3.

The following lemma analyzes the relationship between the Monte Carlo approximations of the VLBs of the $f$-divergence $\mathcal{J}_f$ in Theorem 1 using $\pi(\cdot)$ and $\sigma(\cdot)$ as permutation laws.

**Lemma 3.** *Let $(\mathbf{x}_i, \mathbf{y}_i)$, $\forall i \in \{1, \ldots, N\}$, be $N$ data points. Let $\mathcal{J}_f(T)$ be the value function in (5). Let $\mathcal{J}_f^\pi(T)$ and $\mathcal{J}_f^\sigma(T)$ be numerical implementations of $\mathcal{J}_f(T)$ using a random permutation and a random derangement of $\mathbf{y}$, respectively. Denote with $K$ the number of points $\mathbf{y}_k$, with $k \in \{1, \ldots, N\}$, in the same position after the permutation (i.e., the fixed points). Then*

$$\mathcal{J}_f^\pi(T) \leq \frac{N - K}{N} \mathcal{J}_f^\sigma(T). \tag{17}$$

Lemma 3 practically asserts that the value function $\mathcal{J}_f^\pi(T)$ evaluated via a naive permutation of the data is not a valid VLB of the $f$-divergence, and thus, there is no guarantee on the optimality of the discriminator's output. An interesting mathematical connection can be obtained when studying $\mathcal{J}_f^\pi(T)$ as a sort of variational skew-divergence estimator (Lee & Shin, 2022), but this goes beyond the scope of this paper.

The following theorem states that in the case of the KL divergence, the maximum of $\mathcal{J}_f^\pi(D)$ is attained for a value of the discriminator that is not exactly the density ratio (as it should be from (23), see Appendix A).

**Theorem 2.** *Let the discriminator $D(\cdot)$ be with enough capacity. Let $N$ be the batch size and $f$ be the generator of the KL divergence. Let $\mathcal{J}_{KL}^\pi(D)$ be defined as*

$$\mathcal{J}_{KL}^\pi(D) = \mathbb{E}_{(\mathbf{x},\mathbf{y}) \sim p_{XY}(\mathbf{x},\mathbf{y})} \left[ \log\left( D(\mathbf{x}, \mathbf{y}) \right) - f^*\left( \log\left( D(\mathbf{x}, \pi(\mathbf{y})) \right) \right) \right]. \tag{18}$$

*Denote with $K$ the number of indices in the same position after the permutation (i.e., the fixed points), and with $R(\mathbf{x}, \mathbf{y})$ the density ratio in (2). Then,*

$$\hat{D}(\mathbf{x}, \mathbf{y}) = \arg\max_D \mathcal{J}_{KL}^\pi(D) = \frac{N R(\mathbf{x}, \mathbf{y})}{K R(\mathbf{x}, \mathbf{y}) + N - K}. \tag{19}$$

Although Theorem 2 is stated for the KL divergence, it can be easily extended to any $f$-divergence using Theorem 1. Notice that if the number of indices in the same position $K$ is equal to 0, we fall back into the derangement strategy and we retrieve the density ratio as output.

When we parametrize $D$ with a neural network, we perform multiple training iterations and so we have multiple batches of dimension $N$. This turns into an average analysis on $K$. We report in the Appendix (see Lemma 5) the proof that, on average, $K$ is equal to 1.

From the previous results, it follows immediately that the estimator obtained using a naive permutation strategy is biased and upper bounded by a function of the batch size $N$.

**Corollary 2.1** (Permutation bound). *Let KL-DIME be the estimator obtained via iterative optimization of $\mathcal{J}_{KL}^\pi(D)$, using a batch of size $N$ every training step. Then,*

$$I_{KL-DIME}^\pi := \mathbb{E}_{(\mathbf{x},\mathbf{y}) \sim p_{XY}(\mathbf{x},\mathbf{y})} \left[ \log\left( \hat{D}(\mathbf{x}, \mathbf{y}) \right) \right] < \log(N). \tag{20}$$

We report in Fig. 1 an example of the difference between the derangement and permutation strategies. The estimate attained by using the permutation mechanism, showed in Fig. 1b, demonstrates Theorem 2 and Corollary 2.1, as the upper bound corresponding to $\log(N)$ (with $N = 128$) is clearly visible.

## 6 EXPERIMENTAL RESULTS

In this section, we first describe the architectures of the proposed estimators. Then, we outline the data used to estimate the MI, comment on the performance of the discussed estimators in different scenarios, also analyzing their computational complexity. Finally, we report in Appendix D.2 the self-consistency tests (Song & Ermon, 2020) over image datasets.

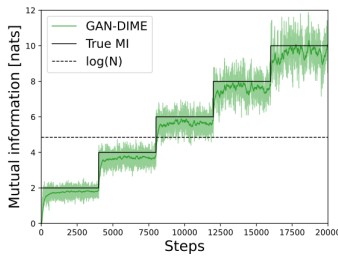
(a) Derangement strategy.

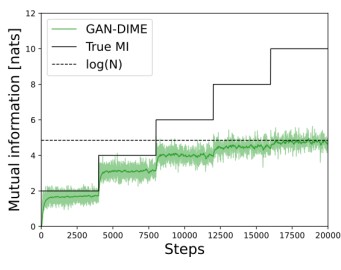
(b) Permutation strategy.

Figure 1: MI estimate obtained with derangement and permutation training procedures, for data dimension $d = 20$ and batch size $N = 128$.

## 6.1 ARCHITECTURES

To demonstrate the behavior of the state-of-the-art MI estimators, we consider multiple neural network *architectures*. The word architecture needs to be intended in a wide-sense, meaning that it represents the neural network architecture and its training strategy. In particular, additionally to the architectures **joint** (Belghazi et al., 2018) and **separable** (Oord et al., 2018), we propose the architecture **deranged**.

The **joint** architecture concatenates the samples $\mathbf{x}$ and $\mathbf{y}$ as input of a single neural network. Each training step requires $N$ realizations $(\mathbf{x}_i, \mathbf{y}_i)$ drawn from $p_{XY}(\mathbf{x}, \mathbf{y})$, for $i \in \{1, \dots, N\}$ and $N(N-1)$ samples $(\mathbf{x}_i, \mathbf{y}_j), \forall i, j \in \{1, \dots, N\}$, with $i \neq j$.

The **separable** architecture comprises two neural networks, the former fed in with $N$ realizations of $X$, the latter with $N$ realizations of $Y$. The inner product between the outputs of the two networks is exploited to obtain the MI estimate.

The proposed **deranged** architecture feeds a neural network with the concatenation of the samples $\mathbf{x}$ and $\mathbf{y}$, similarly to the *joint* architecture. However, the *deranged* one obtains the samples of $p_X(\mathbf{x})p_Y(\mathbf{y})$ by performing a derangement of the realizations $\mathbf{y}$ in the batch sampled from $p_{XY}(\mathbf{x}, \mathbf{y})$. Such diverse training strategy solves the main problem of the *joint* architecture: the difficult scalability to large batch sizes. For large values of $N$, the complexity of the *joint* architecture is $\Omega(N^2)$, while the complexity of the *deranged* one is $\Omega(N)$. NJEE utilizes a specific architecture, in the following referred to as **ad hoc**, comprising $2d-1$ neural networks, where $d$ is the dimension of $X$. $I_{NJEE}$ training procedure is supervised: the input of each neural network does not include the $\mathbf{y}$ samples. All the implementation details are reported in Appendix D.

## 6.2 MULTIVARIATE LINEAR AND NONLINEAR GAUSSIANS

We benchmark the proposed class of MI estimators on two settings utilized in previous papers (Poole et al., 2019; Song & Ermon, 2020). In the first setting (called **Gaussian**), a 20-dimensional Gaussian distribution is sampled to obtain $\mathbf{x}$ and $\mathbf{n}$ samples, independently. Then, $\mathbf{y}$ is obtained as linear combination of $\mathbf{x}$ and $\mathbf{n}$: $\mathbf{y} = \rho \mathbf{x} + \sqrt{1 - \rho^2}\, \mathbf{n}$, where $\rho$ is the correlation coefficient. In the second setting (referred to as **cubic**), the nonlinear transformation $\mathbf{y} \mapsto \mathbf{y}^3$ is applied to the Gaussian samples. The true MI follows a staircase shape, where each step is a multiple of 2 *nats*. Each neural network is trained for 4k iterations for each stair step, with a batch size of 64 samples ($N = 64$). The values $d = 20$ and $N = 64$ are used in the literature to compare MI neural estimators. The tested estimators are: $I_{NJEE}$, $I_{SMILE}$ ($\tau = 1$), $I_{GAN-DIME}$, $I_{HD-DIME}$, $I_{KL-DIME}$, and $I_{CPC}$, as illustrated in Fig. 2. The performance of $I_{MINE}$, $I_{NWJ}$, and $I_{SMILE}(\tau = \infty)$ is reported in Sec. D of the Appendix, since these algorithms exhibit lower performance compared to both SMILE and $f$-DIME. In fact, all the $f$-DIME estimators have lower variance compared to $I_{MINE}$, $I_{NWJ}$, and $I_{SMILE}(\tau = \infty)$, which are characterized by an exponentially increasing variance (see (11)). In particular, all the estimators analyzed belonging to the $f$-DIME class achieve significantly low bias and variance when the true MI is small. Interestingly, for high target MI, different $f$-divergences lead to dissimilar estimation properties. For large MI,

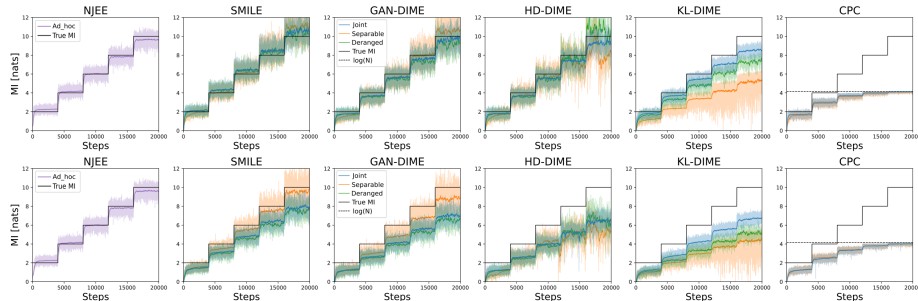

Figure 2: Staircase MI estimation comparison for $d = 20$ and $N = 64$. The *Gaussian* case is reported in the top row, while the *cubic* case is shown in the bottom row.

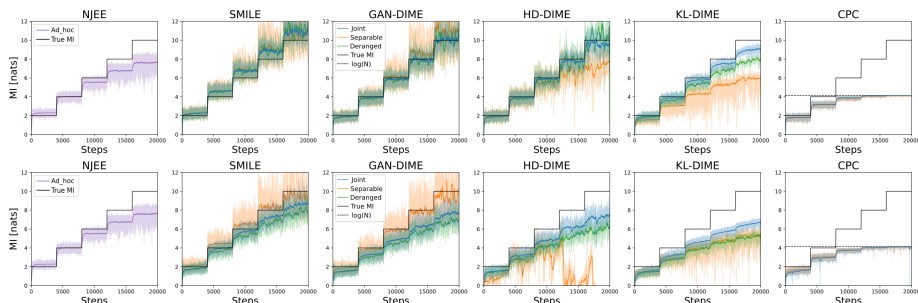

Figure 3: Staircase MI estimation comparison for $d = 5$ and $N = 64$. The *Gaussian* case is reported in the top row, while the *cubic* case is shown in the bottom row.

$I_{KL-DIME}$ is characterized by a low variance, at the expense of a high bias and a slow rise time. Contrarily, $I_{HD-DIME}$ attains a lower bias at the cost of slightly higher variance w.r.t. $I_{KL-DIME}$. Diversely, $I_{GAN-DIME}$ achieves the lowest bias, and a variance comparable to $I_{HD-DIME}$.

The MI estimates obtained with $I_{SMILE}$ and $I_{GAN-DIME}$ appear to possess similar behavior, although the value functions of SMILE and GAN-DIME are structurally different. The reason why $I_{SMILE}$ is almost equivalent to $I_{GAN-DIME}$ resides in their training strategy, since they both minimize the same $f$-divergence. Looking at the implementation code of SMILE [1], in fact, the network's training is guided by the gradient computed using the Jensen-Shannon (JS) divergence (a linear transformation of the GAN divergence). Given the trained network, the clipped objective function proposed in (Song & Ermon, 2020) is only used to compute the MI estimate, since when (31) is used to train the network, the MI estimate diverges (see Fig. 7 in Appendix D). However, with the proposed class of $f$-DIME estimators we show that during the estimation phase the partition function (clipped in (Song & Ermon, 2020)) is not necessary to obtain the MI estimate.

$I_{NJEE}$ obtains an estimate for $d = 20$ and $N = 64$ that has slightly higher bias than $I_{GAN-DIME}$ for large MI values and slightly higher variance than $I_{KL-DIME}$. $I_{CPC}$ is characterized by high bias and low variance. A schematic comparison between all the MI estimators is reported in Tab. 4 in Sec. D of the Appendix.

When $N$ and $d$ vary, the class of $f$-DIME estimators proves its robustness (i.e., maintains low bias and variance), as represented in Fig. 3 and 4. Differently, the behavior of $I_{CPC}$ strongly depends on $N$. At the same time, $I_{NJEE}$ achieves higher bias when $N$ increases and, even more severely, when $d$ decreases (see Fig. 3). Additional results describing all estimators' behavior when $d$ and $N$ vary are reported and described in Appendix D.

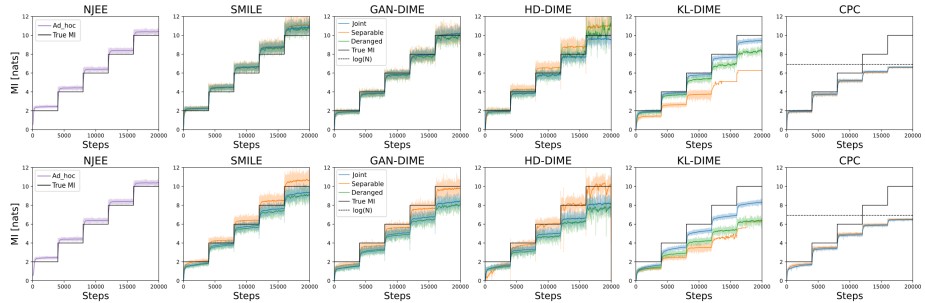

Figure 4: Staircase MI estimation comparison for $d = 20$ and $N = 1024$. The *Gaussian* case is reported in the top row, while the *cubic* case is shown in the bottom row.

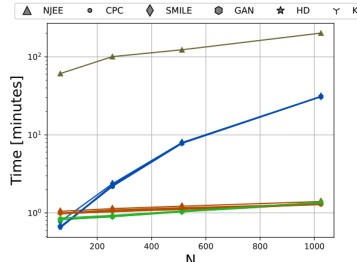 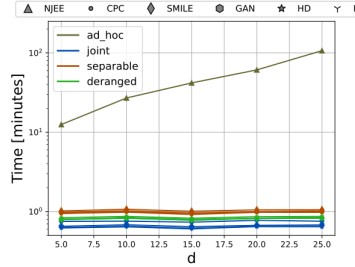

(a) Multivariate Gaussian distribution size fixed to 20. Batch size varying from 64 to 1024.

(b) Multivariate Gaussian distribution size varying from 5 to 25. Batch size fixed to 64.

Figure 5: Time requirements comparison to complete the 5-step staircase MI.

COMPUTATIONAL TIME ANALYSIS

A fundamental characteristic of each algorithm is the computational time. The time requirements to complete the 5-step staircase MI when varying the multivariate Gaussian distribution dimension $d$ and the batch size $N$ are reported in Fig. 5. The difference among the estimators computational time is not significant, when comparing the same architectures. However, as discussed in Sec. 6.1, the *deranged* strategy is significantly faster than the *joint* one as $N$ increases. The fact that the *separable* architecture uses two neural networks implies that when $N$ is significantly large, the *deranged* implementation is much faster than the *separable* one, as well as more stable to the training and distribution parameters, as shown in Appendix D. $I_{NJEE}$ is evaluated with its own architecture, which is the most computationally demanding, because it trains a number of neural networks equal to $2d - 1$. Thus, $I_{NJEE}$ can be utilized only in cases where the time availability is orders of magnitudes higher than the other approaches considered. When $d$ is large, the training of $I_{NJEE}$ fails due to memory requirement problems. For example, our hardware platform (described Appendix D) does not allow the usage of $d > 30$.

## 7 CONCLUSIONS

In this paper, we presented $f$-DIME, a class of discriminative mutual information estimators based on the variational representation of the $f$-divergence. We proved that any valid choice of the function $f$ leads to a low-variance MI estimator which can be parametrized by a neural network. We also proposed a derangement training strategy that efficiently samples from the product of marginal distributions. The performance of $f$-DIME is evaluated using three functions $f$, and it is compared with state-of-the-art estimators. Results demonstrate excellent bias/variance trade-off for different data dimensions and different training parameters.

---

[1]https://github.com/ermongroup/smile-mi-estimator

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
