# OpenReview forum: "Mutual Information Estimation via $f$-Divergence and Data Derangement Based Learning Models"
_ICLR.cc/2024/Conference — Submitted to ICLR 2024_

### Official Review · Reviewer_8P8h · 2023-10-30

**Soundness:** 3 good
**Presentation:** 3 good
**Contribution:** 2 fair
**Rating:** 5
**Confidence:** 3

**Summary:**

This submission presented a solution to mutual information estimation for high-dimensional data using the variational formulation of f-divergence. The author(s) pointed out that naive permutation-based shuffling upper bounds the MI estimate via log(N), and therefore advocate a novel shuffling scheme called derange. The claims are validated using numerical experiments with simple Gaussian variables.

**Strengths:**

The author(s) have presented a novel variational mutual information estimator inspired by f-GAN. The proposed estimator employs a two-stage estimation procedure: (1) optimizing the value function with respect to the critic; (2) estimating the mutual information based on the estimated critic. A key claim of the proposed estimator is that it does not involve any partition function estimation, which is a contributor to extra variance. The author(s) further presented rigorous variance analyses and presented a novel shuffling scheme called derangement to overcome the log(N) cap from vanilla permutation.

**Weaknesses:**

* Inadequate literature coverage and missing baselines. Technically the author(s) are proposing a GAN-like MI estimator, where an f-divergence-based function is optimized to learn the density ratio, then plug in that density ratio estimate into the MI definition to approximate the MI. This is similar in spirit to the DIM work [1], where the JSD objective (also an f-divergence) was used. Also, the idea of dropping the partition function to reduce estimation variance for MI has also been explored in the work of FLO [2], where similar Fenchel duality is applied to cast the log(partition) into a nested optimization problem. These closely related works should be discussed and compared. Also highly relevant is the work of RenyiCL [3], where skewed Renyi divergence is used as a more generic mutual information measure that enjoys bounded variance for numerical estimation.

[1] R. Hjelm, et al. LEARNING DEEP REPRESENTATIONS BY MUTUAL INFORMATION ESTIMATION AND MAXIMIZATION. ICLR 2019
[2] Q Guo, et al. Tight Mutual Information Estimation With Contrastive Fenchel-Legendre Optimization. NeurIPS 2022
[3] K Lee and J Shin. RényiCL: Contrastive Representation Learning with Skew Rényi Divergence. NeurIPS 2022

* Weak experiments. The author(s) only compared different estimators on the toy multi-dimensional Gaussian data, which is inadequate by today's standard. The variational MI estimators are extensively used in self-supervised representation learning (e.g., CLIP, SimCLR, etc.), and the current work failed to provide convincing evidence that the proposed estimator can reliably handle high-dimensional complex data in such applications.
* The statement that "However, Monte Carlo sampling renders MINE a biased estimator. " is inaccurate. The bias comes from Jensen's inequality, not Monte Carlo sampling.
* The statement "Although MINE provides a tighter bound $I_{MINE} ≥ I_{NWJ}$, the NWJ estimator is unbiased." is also misleading. Both MINE and NWJ can be tight estimators. Note that MINE is an approximation to MI, while NWJ is a formal lower bound, so technically you can not compare the two with an inequality.
* The notation of permutation function $\sigma(y)$ is not common in the statistical literature and causes confusion.

**Questions:**

I have trouble understanding the following statements to fully appreciate the proposed solution and dearrangement

* "The partition function estimation $\mathbb{E}_{p_X^N p_Y^N}[\hat{R}]$ represents the major issue when dealing with variational MI estimators"
* "Viceversa, the derangement of y, denoted as $\sigma(y)$, leads to N new random pairs (xi,yj) such that $i \neq j$,$\forall i$ and $j \in {1,··· ,N}$."
* "Denote with K the number of points $y_k$, with $k \in {1, . . . , N }$, in the same position after the permutation (i.e., the fixed points)."

---

> ### Author Response · Authors · 2023-11-18
>
> We thank the Reviewer 8P8H for the extensive and quantitative feedback. In particular, we thank the Reviewer for having spotted the strengths of the paper, it is clear from her/his review that the key points and contributions of the paper have been captured. We provide below our answer to the detailed comments.
>
> **Weaknesses**:
> 1. Despite the fact that [2] and [3] highlighted by the Reviewer are very recent papers, we also came to know about them recently. They are not cited in our paper because they are non-directly connected with our contribution. In particular, it is true that the work in [1] utilizes the JSD (which is only a particular case of our more general formulation) to maximize the MI. But it is also true that for their application,  they do not need to explicitly estimate the MI value, which is instead the purpose of our paper. Moreover, they also rely on Nowozin et. al., 2016 results as we do. Thus, we thought it was enough for us to cite only the latter since this is the starting point.
> For what matters FLO [2], it is true that those authors used the Fenchel duality for deriving their bound (it is very similar to TUBA Poole et. al, 2019 just in a contrastive fashion, and we cited InfoNCE). However, we do not see any connection with our variance analysis as the partition term in FLO naturally remains for the MI estimation (sum over marginal samples in eq. (8) and (9) of [2]).
> Lastly, while the RenyiCL estimator [3] represents a promising approach, especially the skew divergence analysis, it is fundamentally different from our methodology as we extract the critic to compute the MI making the variance analysis in contrast to the VLB (and skew) based ones.
> 2. We understand the importance of such applications, and we indeed demonstrated the effectiveness of our MI estimators in high dimensional scenarios in Fig.11 in the Appendix.
> We want to remark that most of the papers in the literature, including the three cited by the Reviewer, utilize the multi-dimensional Gaussian setup to test their estimators as the main experiment. We studied such scenarios by considering the variation of several important parameters (data dimension, batch size, architecture, $f$-function) to show that our method does not work only on cherry-picked situations. We even studied the time complexity which is often neglected. In addition, we offered the code in a unified and comprehensive manner, allowing for reproducibility, quick comparisons and further analysis. We also ran the self-consistency tests in Sec.C.2 of the Appendix.
> 3. The Reviewer is right and we apologize for the confusion. It is indeed true that the bias is a consequence of the gradients being biased, as the logarithmic term in front of the expectation does not allow averaging of mini-batches for training. Simply switching the logarithmic term with the expectation would violate Jensen’s inequality. We rephrase that sentence with “the logarithm before the expectation in the second term renders MINE a biased estimator.”
> 4. We agree with the Reviewer, although for the same parameters, MINE provides tighter estimates than NWJ (see Sec. 2.2 in (Belghazi et al., 2018)). We can rephrase that sentence as “Although for a fixed $T$ MINE provides a tighter bound (formula), the NWJ estimator is unbiased.”
> 5. We apologize for the confusion. However, as the function $\pi(\cdot)$ was already used before, we decided to use $\sigma(\cdot)$ since it is often used for permutation (see Ch.5 of (Scheinerman, 2012)).
>
> **Questions**:
> 1. That sentence is not directly related to the derangement solution, but rather to the two-stage procedure the Reviewer highlighted in the strengths. What we want to convey is that VLB estimators often exhibit high variance and this is mainly due to the fact that they use the variational representation of the MI also for the estimation. Meanwhile, our second step needs the critic for the MI estimation and not the full value function.
> 2. Short example: a random derangement of $\mathbf{y} = [y_1, y_2, y_3]$ is  $\mathbf{y} = [y_2, y_3, y_1]$ and by definition we ensure that no elements of $\mathbf{y}$ after derangement ends up in the same initial position.
> 3. Fixed points appear only with permutations. In the previous example, a random permutation of $\mathbf{y} = [y_1, y_2, y_3]$ is $[y_2, y_3, y_1]$, and the number of fixed points is 0 as no elements remain in the same position after the permutation. However, another random permutation could be $[y_1, y_3, y_2]$ and in this case $y_1$ is a fixed point, the only one, thus in this example $K=1$.
>
> [Ref] Scheinerman, Edward A. Mathematics: a discrete introduction. Cengage Learning, 2012.
>
> **Rating**: We would like to politely say that we are a little surprised with the Reviewer’s rating since she/he highlighted numerous strengths. We hope that our answers help the Reviewer to clarify the missing details so that to increase the chances for a more positive rating. Thank you.

---

> > ### Author Response · Authors · 2023-11-22
> > **Approaching the end**
> >
> > Dear Reviewer 8P8h,
> >
> > Since the discussion period is approaching its end, we kindly ask again the Reviewer to take a look at our **updated paper** and responses, if possible reevaluating our work. Thanks.
> >
> > The Authors

---

### Official Review · Reviewer_8KDZ · 2023-10-31

**Soundness:** 3 good
**Presentation:** 3 good
**Contribution:** 2 fair
**Rating:** 5
**Confidence:** 4

**Summary:**

This paper extended the mutual information neural estimators (MINE) by using f-divergence in the variational form of mutual information. The variance of the proposed estimator is analyzed in Section4. In section 5, the authors proposed a derangement strategy in order to solve the problem caused by the "fixed points" after permutation. The proposed estimators are tested via high-dimensional linear and nonlinear Gaussian synthetic datasets.

**Strengths:**

This paper extended the MINE using f-divergence. As we know, f-divergence is a wide generalization of many different divergence measures in information theory. So f-MINE can be a very natural but general framework for a family of MINE-like estimators. A more general framework will always be helpful for a deeper understanding of these estimators.

**Weaknesses:**

I think the relation between f-MINE proposed in this paper and previous estimators is still not explained very clearly. Also, it is good to explore more about the bias, variance and the derangement strategy. Please refer to the specific questions listed below. Thanks.

**Questions:**

1. Will f-MINE be degenerated to traditional MINE, or other estimators reviewed in Section 2, by choosing different f? If so, then f-MINE is a more general framework. If not, which estimators can be degenerated from f-MINE and which ones can not? It's good to make these clear in order to connect f-MINE with previous works.
2. Is the variance of f-MINE related to the choices of f in theory? If so, how to choose f to minimize the variance. If not in theory, can we observe the same or similar variance for arbitrary choice of f?
3. What about the bias of f-MINE. When the variance is analyzed, the readers are always curious about the bias.
4. It seems that the experiments in Section 6 are all based on derangement strategy proposed in Section 5. In order to prove the usefulness of derangement, it is good to compare the results without derangement.

---

> ### Author Response · Authors · 2023-11-18
>
> We thank the Reviewer for the valuable feedback. We provide below our answer to the detailed comments.
>
> **Weaknesses**:
> It is correct that our framework utilizes the $f$-divergence to construct a family of MI estimators. However, we do not see such framework as a generalization of MINE for the following reasons:
> 1. MINE is based on the DV representation of the MI and it is structurally different from an $f$-divergence (it cannot be obtained starting from  a particular $f$-divergence)
> 2. MINE and other similar variational lower bounds based estimators (NWJ, SMILE, etc.), utilize the value function to compute the MI, and they suffer from known problems such as high variance, explained in eq. (14) and shown in Fig. 5
> 3. The name of the proposed family of estimators is f-DIME (not f-MINE), and this choice comes from how they work: they use an f-divergence loss function to train a discriminator. At convergence, they use such a discriminator to extract the likelihood and compute the MI.
>
> Bias and variance are widely analyzed in the experimental sections of the main paper (see Sec. 6) and in the Appendix C (see Figs. 7,8,9,10, and Tabs. 1,2,3). Moreover, the variance is also theoretically analyzed in Sec. 4 as the low variance of our estimator constitutes an important strength point. The derangement strategy is also analyzed both in its theoretical section (see Sec. 5) and in the Appendix B and C (see Fig. 12). Indeed, the architecture referred to as “deranged” exploits the derangement technique and we offer comparisons with both other architectures and an architecture that simply uses permutations.
>
> **Questions**:
> 1. As mentioned before, MINE cannot be obtained from $f$-DIME as $f$-DIME utilizes a variational representation of the $f$-divergence for training, meanwhile most of the variational lower bound (VLB) estimators proposed so far in literature utilize a variational representation of the MI for training and inference. One could say that the only case where $f$-DIME coincides with the current VLB estimators is when $f$ is the generator of the KL divergence. But even in such case, our approach uses the variational representation *only* for training, as for the MI estimation we utilize the optimal likelihood ratio (see Th. 1). In conclusion, $f$-DIME is indeed a general framework, but structurally different from the SOTA estimators based on the MI VLB.
>
> 2. The choice of $f$ does have an impact on the variance of the estimator.
> We empirically analyzed such an impact in Sec. 6, please see Figs. 1,2,3 and also in the Appendix, see Figs. 7,8,9,10 and Tables. 1,2.
> From a theoretical point of view, we studied the variance of the Monte Carlo MI estimator, considering the density ratio as given. We could not exempt ourselves from carrying it out as this was also done in (Song & Ermon, 2020). When the optimal density ratio is given, the $f$ does not have an impact on the variance.
> For practical and real scenarios, we did study in Lemma 3 the effect of not reaching optimality in the MI estimator. In particular, the proof of Lemma 3 (see eq.(75) in the Appendix B) shows that a discrepancy in the optimal density ratio (or log-density ratio) reflects into a discrepancy of the instantaneous MI by a factor $\Biggl[\frac{\mathrm{d}}{\mathrm{d}T} \log \bigl( \bigl(f^{*}\bigr)^{\prime}(T)\bigr) \biggr|_{T=\hat{T}} \Biggr]$, where $\hat{T}(x) = f^{\prime} \bigl(\hat{R}(x)\bigr)$, as in eq.(38).
>
> 3. We empirically studied the bias of the f-DIME estimator for different $f$ functions in different scenarios, please see Figs.7,8,9, and Tables. 1,2. Lemma 3 also provided useful mathematical insights about the effect of $f$ on the bias. We decided to focus an entire section on the variance analysis as the variance represents the major issue for discriminative MI estimators (see Sec. 4 of (Song & Ermon, 2020)).
>
> 4. The Reviewer is right to say that it is interesting to compare the architecture based on derangement with the one without. As a matter of fact, we did such a comparison in the Appendix, please see Fig. 12 where we show how the architecture based on simple permutations leads to bounded estimations, in accordance with the main findings of Lemma 2, Th. 2 and Cor. 2.1
>
> **Rating**: We hope that the answers provided clarify the reviewer’s doubts and can contribute to grading our paper more positively. Thank you.

---

> > ### Author Response · Authors · 2023-11-22
> > **Approaching the end**
> >
> > Dear Reviewer 8KDZ,
> >
> > Since the discussion period is approaching its end, we kindly ask again the Reviewer to take a look at our **updated paper** and responses, if possible reevaluating our work. Thanks.
> >
> > The Authors

---

### Official Review · Reviewer_rzBW · 2023-10-31

**Soundness:** 3 good
**Presentation:** 3 good
**Contribution:** 2 fair
**Rating:** 6
**Confidence:** 4

**Summary:**

The paper introduces Discriminative Mutual Information Estimator (f-DIME) as a family of mutual information (MI) estimators based on the relation between the dual representation of f-divergences and the ratio between joint and marginal distributions. A theoretical section first justifies f-DIME by characterizing the variance of the estimators with respect to other popular estimation strategies in the literature, then motivates the use of derangements instead of permutations when training the discriminative ratios. Experimental results confirm that f-DIME results in competitive MI estimation, showcasing the effect of the choice of f-divergence and the use of derangements on simple tasks.

**Strengths:**

1) The paper addresses a relevant problem in the literature, proposing a well-motivated strategy to mitigate the variance of discriminative MI estimators.

2) Claims in the paper are backed up by a solid theoretical analysis.

3) Section 6 includes a meaningful analysis of the computation time as a function of the batch size for various critic architectures.

**Weaknesses:**

1) The experiments included in the main text are quite limited since they consider solely simple unimodal distributions. The reported experimental evidence seems limited to support the claim that f-DIME offers higher accuracy when compared to other estimators in the literature.

2) The paper mentions the similarity between SMILE, GAN-DIME (and JS in [1]), further discussion is required to better underline the differences between the estimators, especially considering that the performance seems hardly distinguishable.

 3) Although the proposed method is novel in the context of mutual information estimation, the paper builds upon a well-established theory on the estimation of f-divergences. The comparison between the use of permutations and derangements to produce samples from $p({\bf x})p({\bf y})$ is also novel although similar methods have been already used in practice in existing implementations.

 4) Many relevant parts of the theory and the experimental results are not included in the main text. I would recommend restructuring the submission by potentially moving detailed descriptions of existing estimators into the appendix.

### References
[1]  Poole, Ben, et al. "On variational bounds of mutual information." International Conference on Machine Learning. PMLR, 2019.

**Questions:**

1) The analysis in section 4 focuses on characterizing the variance of the f-DIME estimator (equation 12), but the convergence of the ratio estimator $T({\bf x},{\bf y})$ is determined by the variance of the training objective in equation 10. Does the training variance for the objective in 10 relate to the sub-optimality (bias) of the estimator?

2) What are the main differences between GAN-DIME/JS and KL-DIME/NWJ? I believe that a more direct experimental comparison could provide additional value to the submission.

3) Is there any rationale behind the choice of $f$-divergence to use for f-DIME? What are the main advantages and downsides of each choice, and how do relate to $f$?

---

> ### Author Response · Authors · 2023-11-18
>
> We thank the Reviewer for the time and the constructive feedback.
>
> **Weaknesses**:
> 1. We used the same experimental setting as the most cited papers in the neural MI estimation domain (Song & Ermon, 2020; Poole et al., 2019), to compare our proposed estimator with the state-of-the-art by using the same benchmark. We studied the performance behavior when varying $d$ and $N$, the consistency tests with image datasets, and the computational time (as stated by the Reviewer in the strengths). We also provide a ready-to-use code that unifies all the discussed estimators in the current paper.
>
> 2. As we wrote in Sec. 6.2 and Appendix C.1.1, SMILE achieves the results shown in Fig. 1,2,3 only when the neural estimator is trained by using the JS divergence (following the idea presented in [1]). Differently, when the NN is trained via eq. (6), the results achieved are worse as reported in Fig. 6 in Appendix C. We were rather surprised to discover that the authors of (Song & Ermon, 2020) did not specify any usage of JS in the original paper although they were using it in their publicly available code. Therefore, the performance between GAN-DIME and SMILE ($\tau=1$) is hardly distinguishable because SMILE uses (6) only to compute the MI estimate, but setting $\tau$ to a small value does not significantly impact the MI estimate obtained by using only the first term in (6), which is exactly the MI estimate attained by using GAN-DIME.
>
> 3. Since the Reviewer recognizes the novelty of the $f$-divergence formulation for MI estimation and the theoretical analysis that we provide on the derangement strategy, we see this more as a strength rather than weakness. In addition, to the best of our knowledge, no paper has neither theoretically analyzed the impact of derangements on neural MI estimators nor provided an upper bound when using random permutations.
>
> 4. We decided to include the previous work in the main text because it helps the reader understand better our novel contribution. However, following the Reviewer suggestion, we will move part of the related work in the Appendix to include Lemma 3 in the main text.
>
> **Questions**:
> 1. The Reviewer is right, as in practical scenarios, there is always a numerical bias coming from sub-optimality. The training variance of eq. (10) has an impact on it, but mostly its derivative as it appears in the gradient descent method. More precisely, in Lemma 3, we study the effect of not reaching optimality in the MI estimator and show that a discrepancy in the optimal density ratio (or log-density ratio) reflects into a discrepancy of the instantaneous MI by a factor $\delta^{(n)} \cdot \Biggl[\frac{\mathrm{d}}{\mathrm{d}T} \log \bigl( \bigl(f^{*}\bigr)^{\prime}(T)\bigr) \biggr|_{T=\hat{T}} \Biggr]$, where $\hat{T}(x) = f^{\prime} \bigl(\hat{R}(x)\bigr)$, as in eq.(38). In particular higher variance in the batch gradient estimates have an impact on $\delta^{(n)}$ and this term is modulated (multiplied) by a factor dependent on $f$.
>
> 2. The major difference between GAN-DIME/JS and KL-DIME/NWJ is the estimator itself, since the class of $f$-DIME estimators does not rely on the estimation of the partition function (as explained in Sec. 4). The difference between GAN-DIME/JS and KL-DIME/NWJ is significant when looking at the graphs in (Poole et al., 2019; Song & Ermon, 2020) and the ones in our paper, because the variance of $f$-DIME is significantly lower. For instance, by comparing Figs. 1 and 5 (achieved using the same parameters) the variances of KL-DIME and GAN-DIME are extremely lower w.r.t. NWJ and SMILE with $\tau=\infty$. Differently, SMILE with $\tau=1$ has a variance comparable to GAN-DIME, showing the importance of the variance reduction that is obtained by reducing the effect of the partition function.
>
> 3. We reported a summary and comparison of the estimators in Appendix C.1.5 and Table 3. In short, in an unknown scenario, we advise to use GAN-DIME, since it is the most robust estimator and achieves excellent performance on all the tested scenarios. If the true MI is known to be low-valued, then KL-DIME is a slightly better choice. However, we also believe that the proposed methodology may stimulate other readers to find new $f$-divergences that may provide performance even higher than GAN-DIME.
>
> **Rating**: We hope that the answers provided clarify the reviewer’s doubts and can contribute to grading our paper more positively. Thank you.

---

> > ### Comment · Reviewer_rzBW · 2023-11-20
> >
> > I would like to express my gratitude to the authors for their time and answers. I appreciate the authors' modification to the submission, and I believe that the overall quality has improved. However, there remain some critical aspects that I feel are not fully addressed.
> >
> >
> > 1) **Experimental results** I recognize that the authors rely on experimental settings that are well-established in the literature. However, I share the concerns raised by reviewers FGDT and 8P8h regarding the experimental section. It currently does not sufficiently highlight the distinct characteristics of the estimators or provide clear insights into the optimal choice of $f$-divergence. Additionally, recent work [1] has pointed out the limitations of using unimodal distributions for benchmarking MI estimators, which further underscores the need for a more comprehensive experimental analysis.
> >
> > 2) **Comparison with JS-like estimators** I thank the authors for their detailed answers however I struggle to understand their point regarding small values of $\tau$. Can the author please clarify what they mean with the sentence "setting $\tau$ to a small value does not significantly impact the MI estimate obtained by using only the first term in (6)"? Which first term are they referring to?
> > Overall, I believe that the submission would benefit from a more direct comparison between the two approaches (JS as in [1] and GAN-DIME) in the main text.
> >
> > 3) **Sub-optimality and Bias** Thank you for the insightful responses related to question 1. However, I am somewhat perplexed by the references to lemmas and equations, as it appears there might be a mix-up between Lemma 1 and 3. The bias expression in equation 52 is crucial for a deeper understanding of the role of $f$ and for characterizing the bias of the estimators. I believe the paper would benefit significantly from both theoretical and empirical expansions in this area. Specifically, an empirical comparison of the variance in (5) and the bias in (7) would effectively illustrate the bias-variance trade-off for the f-DIME estimators.
> >
> >
> > ### References
> > [1] Czyż, Paweł, et al. "Beyond Normal: On the Evaluation of Mutual Information Estimators." arXiv preprint arXiv:2306.11078 (2023).

---

> > > ### Author Response · Authors · 2023-11-20
> > >
> > > We thank the Reviewer for the positive feedback and for having acknowledged that the quality of the updated version is better. However,  we are quite surprised to see that the grade did not increase despite the positive comment of the Reviewer.
> > >
> > > **Experimental results**
> > >
> > > With all due respect, we do not completely agree with the statement of the Reviewer. In fact, we believe that the experiments, despite the established Gaussian setting, actually validated the theoretical results presented in this paper:
> > > - lower variance compared to classical VLB MI estimators thanks to the absence of the partition term during the estimation (see also Fig. 6);
> > > - faster and accurate estimation using the newly proposed derangement training strategy (and its comparison with results obtained via permutations, see Fig. 1 in the updated version);
> > > - excellent bias-variance tradeoff for different dimension $d$ and batch size $N$, as it can be observed from Fig. 2,3,4 and Tables 1,2,3, especially when using GAN-DIME. Indeed, we also discussed which $f$ to use in the majority of scenarios (see Table 4).
> > >
> > > As a matter of fact, despite [1] claiming that using only Gaussian distributions as benchmarks lead to non fully-descriptive results, we did run experiments on multimodal distributions during the evaluation of the self-consistency tests. In particular, we used the MNIST image dataset, known to be multimodal.
> > >
> > > **Comparison with JS-like estimators**
> > >
> > > We thank the Reviewer for having appreciated our detailed answers.
> > > The reference “(6)” was referred to the mathematical formula of $I_{SMILE}$ in the old pdf version, since our answer to Reviewer rzBW was posted on 18 Nov at 16.48h, while we updated the pdf (as the Reviewer noticed) on 19 Nov at 22.55h. The same formula is identified as (31) in Appendix B. We provide a more detailed explanation of the sentence “setting to a small value does not significantly impact the MI estimate obtained by using only the first term in (6)” in the following.
> > >
> > > The “first term” in (31) (old (6)) is the expectation over $p_{XY}(\mathbf{x}, \mathbf{y})$. When $\tau$ is small (e.g., $\tau=1$), $e^{\tau}$ and $e^{-\tau}$ are also small (e.g., $e$ and $\frac{1}{e}$). Since the second term of (31) (old (6)) was proven to be characterized by high-variance in (Song & Ermon, 2020), especially for high MI values,  the clipping function helps in controlling the variance of $I_{SMILE}$, keeping it low. Because of the clipping, for high MI values the second term marginally contributes to the MI estimation.
> > >
> > > We will include the more detailed comparison between GAN-DIME, JS, and SMILE as we reported in the previous answer to the Reviewer. However, we think that the space limitations force us to write such a paragraph in the Appendix.
> > >
> > > **Sub-optimality and bias**
> > >
> > > We are happy to read that the Reviewer found our answer insightful. We believe that the confusion in the references is caused by the motivations explained before. We apologize for that as we thought OpenReview allowed for version tracking. Given the importance of Lemma 1 (old Lemma 3) and based on the Reviewer’s comment, we added a small paragraph that encourages the reader to carefully check the Lemma also in the Appendix as it provides further insights: “The proof … bias”.
> > >
> > > The empirical comparison between the variance of (5) and the bias of (7) goes beyond the scope of this paper as we only use (5) during training and not for the estimation of the MI as (5) is not a VLB on the MI. The bias of (7) is reported in all the Figures and Tables of the paper. We even proposed mathematical insights (via Lemma 1). Sub-optimality is per-se a topic, in which we are also interested, but given the contributions we focused on and the space constraints of this conference, we leave it for further studies.
> > >
> > > **Conclusions**:
> > > In conclusion, we hope that our answers and our updated version of the paper are in line with the Reviewer’s expectation. If so, please consider accepting our paper. Thank you.

---

> > > > ### Comment · Reviewer_rzBW · 2023-11-21
> > > >
> > > > I thank the authors for their clarifications and additional answers.
> > > > Indeed, the confusion regarding the references stems from the versioning (or lack thereof) on OpenReview.
> > > >
> > > > Regarding the experimental section, while I note the inclusion of the multimodal MNIST experiments in the Appendix, my reservations persist, primarily due to these experiments focusing on self-consistency tests rather than a thorough analysis of the bias-variance tradeoff. I understand the challenges posed by space constraints; however, a more in-depth exploration of the relationship between the sub-optimality of equation (5) and the bias in equation (7) would, in my opinion, significantly enrich the analysis and contribute additional novelty to the study.
> > > >
> > > > Despite these reservations, I acknowledge the overall value of your submission and I have decided to increase my evaluation score.

---

> > > > > ### Author Response · Authors · 2023-11-21
> > > > > **Thank you**
> > > > >
> > > > > We deeply thank the Reviewer for the time spent in providing valuable feedback and for the positive recommendation.
> > > > >
> > > > > Sincerely, the Authors.

---

### Official Review · Reviewer_FGDT · 2023-11-01

**Soundness:** 3 good
**Presentation:** 2 fair
**Contribution:** 3 good
**Rating:** 6
**Confidence:** 4

**Summary:**

- The authors provided the $f$-discriminative mutual information estimators (DIME), which is constructed by a training value function whose maximization leads to a given MI estimator and the variational representation of the $f$-divergence.
- They theoretically investigate the effects of the permutation function used to obtain the marginal training samples through the estimation variance analysis.
- Drawing from their theoretical insights, they have proposed a new architectural solution centered around derangements.

**Strengths:**

- The authors revealed that the MI can be estimated by maximizing an objective function with some Fenchel conjugate $f^*$, which is interesting.
- The authors showed that the MI estimator through their proposed objective has a lower variance than that of the existing one.

**Weaknesses:**

I would like to express my sincere respect for all the efforts the authors have invested in this paper. Unfortunately, however, I cannot strongly recommend this paper as an ICLR 2024 accepted paper for the following reasons: (1) discrepancies between claims and content, (2) ambiguous presentation in the important theoretical analysis part, and (3) concerns about the validity of the experimental evaluation of the proposed method. The details are provided below. If there are any misunderstandings, I apologize, and I would appreciate it if you could explain them to me.

## Discrepancies between claims and content
- The authors claimed that their derangement training strategy achieves the unbounded MI estimation. However, I couldn't found out the theory and proof to justify this claim unless I'm misreading. I can only see that they only showed that the permutation bound is bounded by $\log N$. I read the experimental part, but could not find anything to support this claim either. At this stage, the contributions mentioned in the introduction do not align with the content, and I am compelled to express concerns regarding the validity of this research paper.
- Is it possible to overcome the $\log N$ upper bound on the mutual information neural estimation through the maximization of the variational lower bound, perhaps by employing a simple technique such as derangements? The recent paper [1] has shown that, not just CPC, **any distribution-free high-confidence lower bound on mutual information estimated from $N$ samples cannot be larger than $\mathcal{O}(\log N)$**. When I first read the preface of this paper and found the claim that unbounded estimation had been achieved despite such serious statistical limitations, I was both surprised and hopeful. However, I am disappointed to discover that there is no theoretical basis or related discussion to support this assertion.

## Ambiguous presentation in the important theoretical analysis part
- The theoretical analysis in Sections 4 and 5 is the most crucial part of this paper as it provides strong support for the favorable properties of the method proposed in this paper. Nevertheless, the presentation in these sections introduces assumptions, notations, and proof sketches abruptly before delving into the discussion of the results obtained. This made it quite challenging to grasp the intent and content when I first read it. I believe that it would be challenging for readers to comprehend the content on first encounter if the section's opening does not provide an outline of what is to be demonstrated and the results obtained before sharing the proof's details.
- Given that the primary purpose here is to present theoretical results, it would be advisable to organize them as theorems or proposition and provide formal proofs in the main text or in an appendix for clarity and thoroughness (Just to reiterate, I would like to declare that I understand that the proofs for Lemma 1, 2, and other such elements used in the explanatory proofs of each section are provided in the appendices.).
- I believe that readability is a crucial element in effectively presenting the main theoretical claims. Currently, it is necessary to read the content multiple times to understand it to some extent, which can lead to the final intended message becoming unclear. Additionally, I'm concerned about important theoretical aspects being placed in the appendix. For instance, isn't Lemma 3 an important property of the proposed estimator?

## Concerns about the validity of the experimental evaluation of the proposed method
- I also have concerns regarding the validity of interpreting the experimental results.The authors claim that from the experimental results presented in Figure 1, their proposed f-DIME estimator achieves lower variance compared to conventional methods (MINE, NWJ, SMILE). However, at least to me, it doesn't seem like there is a significant difference. I've also examined the experimental results in Fig. 2-5 and the Appendix, but it appears that there are no results provided to substantiate this claim. Even when reviewing Table 1, which provides quantitative evaluations, it is clear that C PC consistently achieves good performance in terms of variance, which contradicts this claim entirely. I hope it's my misinterpretation, but it's quite challenging to accept this claim as valid.
- The precision of the f-DIME estimator doesn't seem to be consistently good overall. However, there's no mention of this, and in fact, it's claimed to be robust to changes in data size and dimension. To me, it appears that NJEE and SMILES are rather stable. For instance, in Fig. 2, the accuracy of HD-DIME w./ separable in the cubic case seems to deteriorate compared to Fig. 1. It's challenging to confirm the proposed method's exceptional robustness based on these experimental results.
- Is it correct to say that the estimator in the "deranged" setting (the green line) is the most crucial among the f-DIME estimators, such as GAN-DIME? It appears that its accuracy is inferior to other methods in many experiments. However, there is no discussion addressing this, which leaves questions about the claim in the introduction that the proposed approach is state-of-the-art.

## MISC
- I recommend adhering to the ICLR format. The authors seem to have mistakenly used a different font style from the original ICLR template. This is evident, for instance, in the title where the font appears distinct. Furthermore, there also seems to be a slightly wider character spacing throughout the document. This could be a candidate for a desk reject.

**Questions:**

In connection with the weaknesses mentioned above, I would like to pose several questions related to the concerns raised. I would appreciate your responses.

- I would greatly appreciate your response to the concerns outlined in the weakness section, with particular attention to the discrepancies between the claims of contributions and the interpretation of experimental results in contrast to the actual findings, as these are the most crucial issues requiring attention.
- In [1], it has been demonstrated that estimation based on MI lower bounds using neural networks faces difficulties in surpassing the log(N) upper bound. Can the authors' proposed method truly overcome this limit? If it is possible, please provide a formal proof. Additionally, it would be valuable to hear the authors' thoughts on the relationship between this research and [1], while citing it in the process.

## Citation
(Note: I am not the author of the following papers)

[1]: D. McAllester and K. Stratos. Formal Limitations on the Measurement of Mutual Information. In AISTATS2020.

================ AFTER REBUTTAL & DISCUSSION ================

The authors have effectively addressed my misunderstandings and concerns. I would like to extend my gratitude to the authors for their thorough and relevant discussion and rebuttal. I have revised the soundness and contribution scores, upgrading them from 2 to 3. Although the presentation still poses a challenge, I believe this paper is highly deserving of acceptance at ICLR. Therefore, I am raising the overall rating to 6.

**Details Of Ethics Concerns:**

I believe that this work does not raise any ethical concerns because it is a methodological study focused on mutual information neural estimation.

---

> ### Author Response · Authors · 2023-11-18
>
> We thank the Reviewer for the time dedicated to carefully read our paper. We always appreciate constructive criticism as in this case and we are happy to clear out the misunderstandings and explain why, politely, the three highlighted weaknesses are the result of a misinterpretation. The comments appear rather strong/harsh. Hopefully, our reply will help the Reviewer to have a more positive opinion.
> Because of the maximum 5000 character limit, we write the answers in two parts.
>
> **Weaknesses**:
> ## **Discrepancies between claims and content**
> We appreciate that the Reviewer was hopeful about the unboundedness of the proposed estimator and we want to assure the Reviewer that there is no trick, as the paper does actually show that it is possible to overcome the $\log N$ upper bound via derangements, as discussed in the paper with both theoretical arguments (Lemma 2, Th. 2 and Cor. 2.1 and their proofs ) and experimental results (Figs 1, 2,3 and Fig.12 in the Appendix).
>
> Firstly, The hypothesis of the main theorem of D. McAllester et. al. do not apply to the proposed estimator ($f$-DIME) with the derangement strategy. The main idea in that theorem is that, given two distributions $p$ and $q$, it is difficult to distinguish samples of $q$ from samples of another skew distribution $\tilde{q} = 1/N p + (1-1/N)q$. Therefore, when we evaluate the KL divergence between $p$ and  $\tilde{q}$, it is immediate to notice that such divergence is upper bounded by $\log N$. However, this is not our case. In fact, let us suppose to have only access to $N$ joint samples $(x,y)\sim p_{XY}$, shuffling $y$ to obtain marginal samples would lead, with high probability (please see Lemma 5 in the Appendix), to one sample $(x_i,y_i)$ which is a joint sample, falling in a similar condition of the one studied by D. McAllester et. al.
> Given that we have access only to $N$ samples, how can we construct marginal samples from joint samples? The answer is derangements, since by definition, it is guaranteed that no $y$ sample will end up in the same position when shuffled, avoiding to fall back into a skew distribution $q$, which would lead to the $\log N$ bound.
>
> Secondly, please note that Lemma 2 provides an inequality between the two value functions, with permutations and with derangements. In particular, we proved that the value function with permutations is upper bounded by the one with derangements. An even further mathematical connection can be obtained when studying the estimator with the permutation strategy as a sort of variational skew-divergence estimator (see Lee and Shin, 2022), but this goes beyond the scope of this paper.
>
> Thirdly, the experimental results do not show the existence of any bound in our case (see Fig. 1, 2, and 3). As a matter of fact, Fig. 12 in the Appendix also shows the boundedness of the estimator when using the permutations instead of the derangements, which is in line with our theoretical result in Corollary 2.1.
>
> Now, based on the Reviewer's comment, we decided to slightly rephrase our contribution claim in the introduction as follows “We study the impact of data derangement for the learning model and propose a novel derangement training strategy that overcomes the $\log N$ upper bound on the mutual information estimation (D. McAllester et. al.), contrarily to what happens when using a random permutation strategy.”
>
> Please also observe that our contribution in this respect is twofold: a) it shows the boundedness of the f-DIME estimator when using permutations to obtain marginal samples; b) it presents a strategy and an architecture to overcome such bound.
>
> In conclusion, with all the respect, we believe that grading the “soundness” with 1 represents a rather unfair rating. Hopefully, with the clarification above the Reviewer agrees with us.
>
> [Ref] Lee, Kyungmin, and Jinwoo Shin. "RényiCL: Contrastive representation learning with skew rényi divergence." Advances in Neural Information Processing Systems 35 (2022): 6463-6477.
>
> ## **Concerns about the validity of the experimental evaluation of the proposed method**
> For what matters the concern about the variance of the proposed estimator, with all the respect, we believe the Reviewer may have misinterpreted the results. We  explain our reasons below:
> The variance analysis (Sec. 4) compares $f$-DIME and the variational lower bound estimators. This result is theoretically proved and experimentally verified, as it can be observed by comparing the behavior of $f$-DIME estimators with MINE, NWJ, and SMILE (with $\tau=\infty$) (Fig. 5). We believe that the Reviewer might have overlooked this figure as the variance difference with the results obtained in Fig. 1 is clearly visible.
>
> (Please continue our answer in the general comment)

---

> ### Comment · Reviewer_FGDT · 2023-11-19
> **Acknowledgement to the authors' rebuttals**
>
> Thank you for taking the time to read my peer review and for your thoughtful response to my concerns.
> Firstly, I would like to extend my sincerest apologies for my misinterpretation of the content in this paper.
> The authors have sincerely and effectively addressed my concerns, and as a result, I am committed to revising the soundness score, increasing it from 1 to 3, along with adjusting the overall rating accordingly.
>
> Here is a summary of my responses to your rebuttals.
> ## About discrepancies between claims and content
>
> - The authors' responses have significantly clarified the interplay between Lemma 2, Theorem 2, and Corollary 2.1 and their connections to McAllester et al. In my personal feeling, the explanations provided in the responses seemed more accessible compared to the current presentation in the paper.
>
> - (This is just a recommendation, but) I would like to suggest that the second, third, and fourth paragraphs of the response addressing "Discrepancies between claims and content" could be incorporated into the main text.
> For instance, these paragraphs could be integrated into the related work section of the Discussion or placed in an appendix.
> Doing so might enhance the overall clarity of the paper and potentially reduce the number of readers who misread the contribution of this paper as I did.
>
> ## About experiments
> - I appreciate the clarification on the experiment's motivation, and I regret having overlooked Figure 12, as it could have helped me correct my misreading. Conversely saying, it might be worthwhile to incorporate Figure 12. into the introduction or the main text if space permits, considering the potential impact of this result as a visually compelling experimental result supporting the theoretical guarantee of the proposed strategy.
>
> - On the other hand, if I may share my personal impression, it appears challenging to distinctly discern the differences in dispersion from Figure 1. When it comes to claim some improvement, I would like to know the numerical comparison results. Would Table 1 or 2 correspond to the answer to this comment? Or is it possible to provide numerical variances for each of the methods shown in Fig.1? I feel that improving the persuasiveness of "In fact, all..., ... is small" on p. 7 would improve the readability of this paper.
> Enhancing the persuasiveness of the statement "In fact, all..., ... is small" on page 7 could significantly contribute to improving the overall readability of the paper.
>
> ## About ambiguous presentation
> - Whether the current presentation aligns with common practices in ICLR or not, I concur with your perspective that the choice of presentation style and readability is often subjective, varying from individual to individual is a matter of personal preference.
>
> - My primary concern is that the diverse assumptions and definitions introduced in the theoretical statements in this paper make it difficult, at least from my perspective, to grasp them in a unified manner. This gives me the impression that this may make it difficult to effectively communicate the central message derived from your theoretical results.
>
> - Moreover, Section 5 introduces the proposed derangement strategy. However, its explanation is simply confined to the second half of the second paragraph, and I find myself lacking a comprehensive understanding of why the authors anticipated this method to be effective in improving $\log N$ orders. While I could comprehend the theoretical effectiveness of the proposed method as a result, the motivation behind the derangement strategy remains somewhat unclear. This ambiguity might be attributed to my own knowledge limitations, but I believe addressing this aspect would contribute to the overall clarity of this section.
>
> - Finally, I'm pleased to note your consideration of including Lemma 3 in the text. The convergence of the f-DIME framework is an excellent property, and I believe that incorporating the theoretical analysis of f-DIME will enhance the understanding of the significance of this paper in the context of improved training strategies.
>
> ## MISC
> - After Theorem 2. (p.6): Although the Theorem - > Although the Theorem “2”...?
> - It seems that the authors can upload a revised manuscript. If you resubmit a version that reflects the corrections based on the review, I think that it will be easier for all of reviewers to understand how you have improved your paper.
>
> ## Conclusion
> I express sincere gratitude to the authors for clarifying my misconceptions about the content and interpretation of this paper.
> While I acknowledge that my subjective preferences regarding presentation and readability may have reflected some aspects, I hope my feedback will be helpful in making this paper even better.
>
> Sincerely,
> Reviewer FGDT

---

> > ### Author Response · Authors · 2023-11-19
> > **We updated our paper based on Reviewers comments**
> >
> > We deeply thank you for the prompt and precise response. We are happy to read that our replies clarified the majority of the initial concerns. Once again, we appreciate constructive criticism and we are willing to address also the remaining comments with the purpose of improving the quality and readability of our paper:
> >
> > **About discrepancies between claims and content**
> > - We are glad to know that our answer clarified the existing connection among the theoretical results. Based on your feedback, we decided to update our manuscript including parts of the previous answer in order to facilitate better reading and comprehension. Thanks for this valuable feedback.
> > - We thank the Reviewer for the suggestion. Please find an updated version of our manuscript where we included the 3 paragraphs. In particular, we decided to insert the connection with the skew-divergence in the derangement section, we decided to include Fig.12 in the main text (see Sec.5) as also other Reviewers have noted the importance of this figure, and we modified our claim in the introduction following your comment. We hope the new version improves readability and reduces possibilities of confusion.
> >
> > **About experiments**
> > - This is probably our fault as Fig.12 (now Fig. 1) is indeed extremely important for better understanding derangement. Thus, based also on you and other Reviewers’ comments, we decided to include it directly at the end of Sec. 5. In this way, we believe the reader will immediately grasp why derangements are so important and what is the effect of not using them for training.
> > - Thanks for sharing with us your impression. We agree that the results of Fig. 1 (now Fig. 2) might not seem different at a first glance. However, they look convincing (at least in terms of variance, as mentioned in Sec. 6.2) when compared with the estimators in Fig. 5 (now 6). Please note that we started from this setup as this is the setup used also in other established works (see Song & Ermon, 2020; Poole et al., 2019). Thus, it is fair to offer the comparison in the same scenario.
> > Nonetheless, Fig. 2 (now Fig. 3) and 3 (now Fig. 4) are there on purpose to show to the reader that in non cherry-picked scenarios, for either different dimension $d$ or different batch size $N$, $f$-DIME is more robust and exhibits excellent performance, as observed in Sec. 6. To precisely answer the Reviewer and for completeness, we also included the associated Table of Fig.1 (now Fig. 2), with the exact numerical values. See Table 3 in the Appendix.
> >
> > **About ambiguous presentation**
> > - Thank you.
> > - Considering that the primary interest is the effective and precise communication of our findings, we updated the manuscript in this direction. In particular, given the 9 pages constraint (very little), we decided to add another section in the Appendix where we explain with some examples, for the more interested reader, the idea behind the derangement (which is probably difficult to grasp the first time).
> > - Based on your comment, we also added at the beginning of Sec. 5 the intuition behind the derangement idea and why such a strategy could be effective in reducing the $\log N$ bound. We decided to incorporate part of our previous reply as we believe it really provides added value.
> > - We deeply thank the Reviewer for allowing us to reconsider the relevance of Lemma 3. As promised, we added this Lemma in the main text and we also made a small comment explaining why it is important. Please find it (new Lemma 1) in the updated version of the paper.
> >
> > **MISC**
> > - Yes correct, thanks for noticing this. We corrected it in the updated version.
> > - We totally agree with the Reviewer. Please find in the system an updated version of the manuscript that reflects the Reviewers’ comments.
> >
> > **Conclusion**
> > We take again the opportunity to deeply thank the Reviewer for her/his comments. We believe they helped us improve the paper, especially in terms of readability.
> > Lastly, If the Reviewer believes there are still unclear aspects, we are willing to address them as soon as possible, we felt that modifying some of the presentation of the paper was an aspect needed to render it even more convincing. We did it, and we now hope that the opinion/grading of the paper can be higher, hopefully increasing the chances to have it accepted. Thank you.

---

> > > ### Comment · Reviewer_FGDT · 2023-11-22
> > > **Acknowledgement**
> > >
> > > Thank you for the constructive discussion, corrections, and responses to our concerns.
> > > In my opinion, after these modifications, the soundness and contribution are clear in the current manuscript.
> > > I am raising the overall rating to 6.
> > > I wish the authors the best of luck.
> > >
> > > Sincerely,
> > >
> > > Reviewer FGDT

---

> > > > ### Author Response · Authors · 2023-11-22
> > > > **Thank you**
> > > >
> > > > We would like to express our gratitude for your in-depth feedback and constructive discussion which has been incredibly valuable for our submission.
> > > >
> > > > Once again, thank you.
> > > >
> > > > Sincerely, the Authors

---

### Official Review · Reviewer_zX7N · 2023-11-06

**Soundness:** 3 good
**Presentation:** 2 fair
**Contribution:** 2 fair
**Rating:** 6
**Confidence:** 3

**Summary:**

This paper suggests using the variational representation of $f$-divergence to estimate mutual information (MI), the general idea is to extract a density ratio directly from the discriminator output, which is slighly from existing neural network-based MI estimators that optimize a variational bound of MI. Theoretically, authors show that the new estimator has smaller variance. Practically, authors evalute the performances of three different f-divergence MI estimators on benchmark synthetic data and sanity self-consistency tests.

**Strengths:**

1. Using $f$-divergence to estimate MI is new to me.
2. It is good to show that the new estimator has smaller variance.
3. Authors also contribute a new way to sample data from the product of marginal distributions $p(x)p(y)$ by derangement; which theoretically should perform better than random permutation and is always overlooked in previous literature.

**Weaknesses:**

My main concern comes from the claim that the new estimator has best or excellent bias/variance trade-off, which is not clear to me.
In fact, according to Figs. 1 and 2, the HD-DIME and KL-DIME do not perform that well, which usually suffer from large bias than NJEE and SMILE. Moreover, it seems there is no consistent winner for all f-divergence MI estimators that always performs better than others.
For example, when the true MI is large, KL−DIME has a low variance but a high bias; whereas GAN-DIME has low bias but high variance.
Hence, it is hard to say which estimator really achieves a good bias-variance tradeoff; given that some other estimators also perform well, e.g., NJEE.

**Questions:**

1. Are there ways to give more insights on the bias of the new estimator,
especially demonstrating why it has smaller bias when the true MI value is also small?
2. Please refer to weakness
3. Given the inconsistent performance of different $f$-divergence MI estimator, how can we choose the suitable one in practice?

---

> ### Author Response · Authors · 2023-11-18
>
> We thank the Reviewer for the positive feedback. We particularly appreciate that the Reviewer caught the key points of our contribution and highlighted the novelty and positive aspects. Regarding the questions, please find below our point-to-point response.
>
> **Questions**:
> 1. The bias shown in all the figures is a consequence of the non-optimal convergence of the density ratio estimate to the true density ratio while trying to maximize (10).  Lemma 3 studies the effect of this discrepancy and shows that the convergence of $f$-DIME depends on the choice of the $f$-divergence. In particular, from eq. (75) it can be observed that the discrepancy between the true instantaneous MI and the estimated one is controlled by the product between $\delta^{(n)}$ and $\Biggl[\frac{\mathrm{d}}{\mathrm{d}T} \log \bigl( \bigl(f^{*}\bigr)^{\prime}(T)\bigr) \biggr|_{T=\hat{T}} \Biggr]$, where $\hat{T}(x) = f^{\prime} \bigl(\hat{R}(x)\bigr)$. Thus, the latter factor given a certain $f$ is the one to be analyzed since it impacts the gradient descent method. However, since our estimator does not provide a lower bound on the real MI (differently from the others), it cannot be asserted with certainty that the bias decreases with lower MI value, as this does not appear to be always the case (see Fig.1 for GAN-DIME or Fig.11 for HD-DIME).
>
> 2. We have proposed a class of estimators, for which we studied the properties that hold for any choice of $f$-divergence. It is true, as the Reviewer states, that HD-DIME and KL-DIME do not achieve excellent performance in all the scenarios, but the proposed GAN-DIME does. In fact, among all the considered estimators, GAN-DIME attains better or, in few cases, comparable results. Moreover, GAN-DIME is more robust w.r.t. NJEE. In fact, when the latent dimension of the input distribution or the batch size vary, we show in Fig. 2,3 and Tables 1,2 that the performance of NJEE significantly worsens, while the performance of GAN-DIME does not. In addition, as we explain in Sec. 6.2, as the latent dimension increases, the time complexity of NJEE significantly increases, making it an infeasible MI estimator. Differently, GAN-DIME is extremely fast, when using the deranged or separable architectures. The performance of SMILE and GAN-DIME is similar since, as we state in Sec. 6.2 and Appendix C.1.1, SMILE is trained with a JS maximization trick that is equivalent to the GAN-DIME training objective (this is not mentioned in the original paper).
>
> 3. In an unknown scenario, we recommend the usage of GAN-DIME (as discussed in Appendix C.1.5). The rationale behind this choice is that it achieves excellent performance (in terms of bias and variance) for both low and high values of true MI (see Tables 1 and 2 in Appendix C). Table 3 provides a good overview of the proposed estimators strengths, suggesting which ones can be used for low or high MI and which one can be scaled for high dimensions (See summary in Sec. C.1.5 of the Appendix).
>
> **Rating**:
> We hope that our answers clarify the Reviewer’s doubts. In particular, after seeing the positive rating assigned, together with the encouraging feedback on the paper’s contribution, we hope that our answers contribute to increase the confidence of the Reviewer in grading our paper even more positively. Thank you.

---

> > ### Author Response · Authors · 2023-11-22
> > **Approaching the end**
> >
> > Dear Reviewer zX7N,
> >
> > Since the discussion period is approaching its end, we kindly ask again the Reviewer to take a look at our **updated paper** and responses, if possible reevaluating our work. Thanks.
> >
> > The Authors

---

### Author Response · Authors · 2023-11-18
**We thank the Reviewers**

We thank all the Reviewers for taking the time to read our paper and provide their constructive feedback. We appreciate all the effort that was visibly put into the reviewing process. We hope that our answers clarify all the doubts and help to judge our contribution even more positively.

***
**Continuation to Reviewer FGDT reply**

[..] SMILE with $\tau=1$ has a variance similar to GAN-DIME, since a small $\tau$ almost cancels the effect of the partition term. Notice that SMILE is trained by using a JS approach (see Sec. 6). However, for higher values of $\tau$ the variance of SMILE increases (see Fig. 5 and (Song & Ermon, 2020)).

It is true that CPC achieves lowest variance, but at the expense of a large bias as we remarked in Sec. 6. In fact, the MI estimate of CPC is upper bounded by $\log N$. Thus, CPC does not provide a reliable estimate.

For what matters the robustness, we refer to it as the performance variability depending on a change in the parameters $d$ and $N$. NJEE is significantly dependent on $d$, both in terms of bias and time complexity. SMILE is stable because it is obtained with the same objective function used for training GAN-DIME (Even if it is not mentioned in Song & Ermon, 2020). Thus, GAN-DIME is as stable as SMILE, with the advantage of using derangements as training strategy.

For what matters the relevance of the proposed deranged architecture, it increases as the batch size grows. In particular, the deranged architecture is the fastest (see Fig. 4a and 13), which is a fundamental characteristic when the MI estimator must be updated online. For this reason, the deranged architecture may not always exhibit the best performance (see Fig. 11) as it uses far less samples than the other architectures (see Sec. 6.1). In fact, the deranged architecture extracts $N$ marginal samples from $N$ joint samples with the derangement strategy illustrated in Sec. 5., while other architectures such as the joint one, utilize $N*(N-1)$ marginal samples, increasing the precision and reducing the variance, at the expenses of higher complexity.  We discussed these details in the Appendix C.

In summary, the deranged architecture represents an excellent tradeoff between precision and time complexity.

## **Ambiguous presentation in the important theoretical analysis part**
We do respect the Reviewer point of view and we agree that readability is an important aspect and shall not be overlooked also in a paper like ours which is rather mathematical. But we also believe that when it comes to the style of a paper and the way the contribution is structured we enter into a world of personal taste. Our paper follows the style of many other papers published at ICLR. We structured the theoretical results as lemmas, theorems and corollaries. On the other side, we connected all the theorems, lemmas and corollaries through a text description, to ease reading and provide the paper with continuity. All the proofs are reported in Appendix B, which is a common writing practice in ICLR papers. For maximum clarity, in every section, we firstly introduce the content of the section to help the reader understand what the subject of the section is. As a proof of this, in the first rows of Section 4, we state “In the following, we prove that under the previous assumptions, $f$-DIME exhibits better performance in terms of variance w.r.t. some variational estimators with a discriminative approach, e.g., MINE and NWJ”. In the first rows of Sec. 5, we state “we now prove that a naive permutation over the realizations of $Y$ results in an incorrect variational lower bound of the $f$-divergence, causing the MI estimator to be bounded by $\log N$, where $N$ is the batch size. To solve this issue, we propose a derangement strategy.”
If the Reviewer thinks there are some important results not structured as theorem/proof, we ask her/him to explicitly indicate them, as we are willing to rethink how to present them better if necessary.

Regarding Lemma 3, we agree with the Reviewer that it is indeed an important result. This Lemma is currently in the Appendix. To address the Reviewer concern, we can include it in the main text by shrinking the related work description and moving some equations in the Appendix.
We sincerely hope, given our point of view, that the Reviewer changes his opinion on this aspect.

## **MISC**:
We thank the Reviewer for noticing that. Indeed, we accidentally loaded also the fontenc package. We now remove it, thanks.

**Questions**:
1. We answered Question 1 in the reply to the weakness section.
2. We answered Question 2 in the reply to the weakness section. We hope the Reviewer is now convinced about overcoming the $\log N$ limit thanks to the derangements.

**Rating**: We hope that our replies clarify the doubts of the Reviewer and encourage her/him to reconsider the rating, especially if the Reviewer recognizes the novelty of the contribution and the correctness of the mathematical proofs. Thank you.

---

> ### Author Response · Authors · 2023-11-21
> **We updated our paper**
>
> Dear Reviewers,
>
> Thank you again for your feedback. We updated our paper on Sunday so if you find any inconsistencies with lemmas and equation numbers, please check them with the first version of the paper as it seems OpenReview does not allow version tracking.
>
> As the rebuttal deadline approaches, we would really appreciate if you could check our replies and reconsider your mark in light of the latest explanations and additions provided. We remain available to discuss further if needed.

---

### Meta-Review · Area_Chair_tCFF · 2023-12-09

**Metareview:**

In this paper, the authors propose to use an f-divergence to estimate the mutual information, and they propose a new deranged architecture for the neural network that feeds the estimate. The reviewers and the authors have had a comprehensive discussion to address their many comments which have led to a significantly different manuscript from the original submitted. The reviewers see the merits of the proposed paper they still mentioned that the experiments were not comprehensive and that the differences are not well characterized. In the paper, the authors mentioned that their model has a lower variance, but they do not report a number in the whole experimental section. From the experimental section, it is impossible to understand the difference between the proposed mechanisms. The paper still needs considerable work before it can be presented at a major conference.

**Justification For Why Not Higher Score:**

The paper is very unclear and the experimental section is non-scientific or reproducible. The paper might have some merit for a future venue.

**Justification For Why Not Lower Score:**

Not posible.

---

### Decision · Program_Chairs · 2024-01-16

Reject